# Jorunnamycin A Suppresses Stem-Like Phenotypes and Sensitizes Cisplatin-Induced Apoptosis in Cancer Stem-Like Cell-Enriched Spheroids of Human Lung Cancer Cells

**DOI:** 10.3390/md19050261

**Published:** 2021-05-03

**Authors:** Somruethai Sumkhemthong, Supakarn Chamni, Gea U. Ecoy, Pornchanok Taweecheep, Khanit Suwanborirux, Eakachai Prompetchara, Pithi Chanvorachote, Chatchai Chaotham

**Affiliations:** 1Department of Biochemistry and Microbiology, Faculty of Pharmaceutical Sciences, Chulalongkorn University, Bangkok 10330, Thailand; Somruethai.Su@student.chula.ac.th (S.S.); ecoygea@gmail.com (G.U.E.); p.taweecheep@gmail.com (P.T.); 2Department of Pharmacognosy and Pharmaceutical Botany, Faculty of Pharmaceutical Sciences, Chulalongkorn University, Bangkok 10330, Thailand; supakarn.c@pharm.chula.ac.th (S.C.); Khanit.S@pharm.chula.ac.th (K.S.); 3Natural Products and Nanoparticles Research Unit (NP2), Chulalongkorn University, Bangkok 10330, Thailand; 4Department of Pharmacy, School of Health Care Professions, University of San Carlos, Cebu 6000, Philippines; 5Department of Laboratory Medicine, Faculty of Medicine, Chulalongkorn University, Bangkok 10330, Thailand; eakachai.p@chula.ac.th; 6Center of Excellence in Vaccine Research and Development (Chula Vaccine Research Center-Chula VRC), Faculty of Medicine, Chulalongkorn University, Bangkok 10330, Thailand; 7Department of Pharmacology and Physiology, Faculty of Pharmaceutical Sciences, Chulalongkorn University, Bangkok 10330, Thailand; pithi_chan@yahoo.com; 8Cell-Based Drug and Health Products Development Research Unit, Faculty of Pharmaceutical Sciences, Chulalongkorn University, Bangkok 10330, Thailand

**Keywords:** cancer stem-like cells, cisplatin, lung cancer, jorunnamycin A, stemness transcription factors, β-catenin

## Abstract

It has been recognized that cancer stem-like cells (CSCs) in tumor tissue crucially contribute to therapeutic failure, resulting in a high mortality rate in lung cancer patients. Due to their stem-like features of self-renewal and tumor formation, CSCs can lead to drug resistance and tumor recurrence. Herein, the suppressive effect of jorunnamycin A, a bistetrahydroisoquinolinequinone isolated from Thai blue sponge *Xestospongia* sp., on cancer spheroid initiation and self-renewal in the CSCs of human lung cancer cells is revealed. The depletion of stemness transcription factors, including Nanog, Oct-4, and Sox2 in the lung CSC-enriched population treated with jorunnamycin A (0.5 μM), resulted from the activation of GSK-3β and the consequent downregulation of β-catenin. Interestingly, pretreatment with jorunnamycin A at 0.5 μM for 24 h considerably sensitized lung CSCs to cisplatin-induced apoptosis, as evidenced by upregulated p53 and decreased Bcl-2 in jorunnamycin A-pretreated CSC-enriched spheroids. Moreover, the combination treatment of jorunnamycin A (0.5 μM) and cisplatin (25 μM) also diminished CD133-overexpresssing cells presented in CSC-enriched spheroids. Thus, evidence on the regulatory functions of jorunnamycin A may facilitate the development of this marine-derived compound as a novel chemotherapy agent that targets CSCs in lung cancer treatment.

## 1. Introduction

Due to aggressive features and high prevalence rate, lung cancer accounts for most of the cancer deaths worldwide [1]. Although various treatment regimens have been developed, the five-year survival rate of lung cancer patients in all diagnosed stages has not remarkably improved [2]. Tumor heterogeneity presents a major challenge, as proven by subpopulations of cancer stem-like cells (CSCs), which possess the stemness features of self-renewal and differentiation and promote drug resistance, metastasis, tumorigenicity, and cancer relapse [3,4,5,6,7,8]. Although cisplatin-based chemotherapy remains the first-line drug for lung cancer treatment, eventual therapeutic failure and selection of multidrug-resistant CSC subpopulations are major issues [9,10,11]. Meanwhile, cancer immunotherapy has made remarkable progress in the last decade, but there are persisting limitations such as the low percentage of responders, side effects, and cost-effectiveness [12,13]. At this point, there is a continued need to develop effective CSC-targeted therapy that could thus improve the current standard of care.

Clinical specimens from lung cancer patients have been found to highly express CD133 (Prominin-1), a recognized CSC protein marker in various cancers [14,15,16]. The upregulation of CD133 has allowed CSC tracking and isolation [17]. The CD133high cancer subpopulation demonstrates self-renewal and tumorigenicity in both in vitro and in vivo experiments [18,19]. Furthermore, the CSC phenotype is linked to poor clinical outcomes and therapeutic resistance. The phenotype is correlated with the overexpression of stemness-regulating transcription factors Oct-4 (Octamer-binding transcription factor 4), Sox2 ((Sex determining region Y)-box 2), and Nanog [20]. The decreased sensitivity to apoptosis induction in CD133-overexpressing CSCs may be due to the suppression of tumor suppressor p53 protein by Nanog as well as to the elevation of Bcl-2 (B-cell lymphoma 2) protein [21,22,23]. The regulatory role of the PI3K (phosphatidylinositol-3-kinase)/Akt (protein kinase B)/β-catenin pathway in modulating the CSC phenotype is a point of interest [24]. The expression of self-renewal transcription factors in lung CSCs is mediated by Akt, an up-stream signaling molecule [25,26,27]. After phosphorylation by activated Akt (p-Akt), GSK-3β (glycogen synthase kinase-3β) will release β-catenin from the degradation complex. Sequestered β-catenin then translocates into the nucleus to stimulate the transcription of target genes, including Nanog and Oct-4 [28]. The suppression of Akt and p-GSK-3β has been remarkably noted to suppress CSCs in lung cancer [26].

Jorunnamycin A, a bistetrahydroisoquinolinequinone that has been extracted from Thai blue sponge *Xestospongia* sp. for this study, is also found in the Thai nudibranch *Jorunna funebris* [29]. This natural marine product is similar to ecteinascidin 743 [30] and to renieramycin M, with the key difference being the side chain at the C-22 position (Figure 1), as renieramycin M possesses an angeloyl ester while jorunnamycin A has an alcohol side chain instead. Jorunnamycin A and renieramycin M derivatives have exhibited potent cytotoxicity against human lung cancer cells [31,32,33]. A recent study revealed the inhibitory effect of renieramycin M on CSC features of lung cancer cells; however, its regulatory mechanism has never been investigated [34]. Interestingly, jorunnamycin A was found to promote detachment-induced cell death and to inhibit epithelial-to-mesenchymal transition (EMT) and anchorage-independent growth in human lung cancers [35]. This study aimed to investigate the suppressive activity of jorunnamycin A on stem-like phenotypes of CSCs from human lung cancer cells and its underlying mechanisms. The findings may facilitate the development of this marine-derived compound as a novel chemotherapy for targeting CSCs in lung cancer treatment, which could also serve as an adjuvant that improves the current standard of care.

## 2. Results

### 2.1. Jorunnamycin A Diminishes Cancer Spheroid-Initiating Cells in Lung Cancer H460 Cells

According to the cytotoxic profile obtained from a previous study [35], jorunnamycin A at a nontoxic concentration range between 0.05–0.5 μM was chosen for the evaluation of its targeting effect against the human lung CSC subpopulation. Herein, the half-maximal concentration that reduced 50% cell viability (IC_50_) and inhibited 50% growth (IG_50_) of jorunnamycin A in both lung cancer (H460, H23, and A549) and bronchial epithelial cells (BEAS-2B) was also determined and is presented in Table 1. Notably, the selective anticancer activity of jorunnamycin A was evidenced with higher IC_50_ and IG_50_ value in BEAS-2B cells compared with various lung cancer cells.

As H460 has been reported to not only being composed of a high number of CSCs but also having aggressive self-renewal property and in vivo tumorigenicity [25,26,27,36], the effect of jorunnamycin A targeting on the CSC subpopulation was primarily evaluated in human lung cancer H460 cells. To clarify the inhibitory effect of jorunnamycin A on CSCs, a limiting dilution assay (LDA) was used to quantify the frequency of tumor-initiating cells in cancer populations. After culture for 14 days under detachment condition and deprivation of nutrients and growth factors in LDA, the formation of cancer spheroids mostly resulted from CSCs that possessed aggressive features and tumor-initiating activity [37]. As demonstrated in Figure 2A, the formation of cancer spheroids in LDA at day 14 was evident in human lung cancer H460 cells. It should be noted that H460 cells at different cell densities (1–200 cells/well in ultra-low attachment 96 well-plate) strongly showed spheroid-forming activity, which suggests the CSC phenotype. Intriguingly, jorunnamycin A at nontoxic concentrations (0.05–0.5 µM) significantly diminished colony formation in H460 cells at all cell densities used. To confirm anticancer activity, cisplatin, a recommended chemotherapy agent for lung cancer treatment, was used at toxic concentration as a positive control in this study. Both jorunnamycin A (0.5 µM) and cisplatin (25 µM) comparably reduced colony number in lung cancer H460 cells (Figure 2B). These results suggest that jorunnamycin A could diminish the frequency of cancer spheroid-initiating cells in the CSC subpopulation of human lung cancer cells.

### 2.2. Suppressive Effect of Jorunnamycin A in CSC-Enriched Lung Cancer Cells

Successful assays for determining treatment effects on H460 CSCs involve with the enrichment of the CSC subpopulation and investigation of the stemness traits of self-renewal and differentiation in CSC-enriched spheroids cultured under detachment condition [26]. The formation of CSC-enriched spheroids derived from lung cancer H460 cells was assessed after culture with 0–0.5 μM jorunnamycin A from days 0–7. Figure 2C depicts the anchorage-independent growth of CSC-enriched H460 spheroids after culture for 7 days. Although the incubation with either jorunnamycin A (0.05–0.5 μM) or cisplatin (25 μM) obviously suppressed the enlargement of CSC spheroids, no apoptosis was visibly revealed in Hoechst33342 staining. Importantly, the relative spheroid size was significantly reduced in jorunnamycin A-treated CSC spheroids in a time- and dose-dependent manner (Figure 2D).

As CD133 is a selective marker for lung CSCs [11], the detection of CD133 in CSC-enriched H460 spheroids was performed to further assess the self-renewal trait. Thus, the alteration of the CD133low and CD133high subpopulations in CSC-enriched spheroids after 3 days treatment was evaluated via flow cytometry, and the results are presented in dot plots (Figure 2E). Figure 2F reveals that approximately 80% of the cells containing H460 spheroids were CD133-overexpressing cells, which confirms the CSC-enrichment of spheroids. Both jorunnamycin A (0.1–0.5 μM) and cisplatin (25 μM) treatment significantly decreased the CD133high population compared with the untreated CSC spheroids. The treatments also evidently increased the %CD133low population in CSC-enriched H460 spheroids.

### 2.3. Jorunnamycin A Downregulates Stemness Transcription Factors and Related Proteins in CSC-Enriched Spheroids

The modulation on transcription factors involved in the stemness phenotype was further elucidated in the CSC-enriched lung cancer cells cultured with jorunnamycin A. The mRNA level of Nanog, Oct-4, and Sox2 determined by reverse transcription quantitative real-time PCR (RT-qPCR) is indicated in Figure 3A. Jorunnamycin A at 0.1–0.5 µM dramatically downregulated mRNA level of Nanog, Oct-4, and Sox2 in CSC H460 spheroids promptly at 24 h of treatment. The dose-dependent decrease in the protein level of Nanog and Oct-4 detected via Western blot analysis was also observed in the CSC-enriched spheroids cultured with jorunnamycin A for 24 h (Figure 3B). However, the significant reduction of Sox2 protein level was only observed in CSC H460 spheroids in response to treatment with 0.5 µM jorunnamycin A (Figure 3C). It is worth noting that the alteration of these stemness transcription factors correlated with the lower relative spheroid size observed at day 1 after jorunnamycin A treatment, compared with the non-treated control.

Because the PI3K/Akt/β-catenin pathway has been reported to be an up-stream signal regulating CSC phenotype [24], CSC-enriched H460 spheroids were treated with jorunnamycin A (0–0.5 μM) for 24 h to evaluate the expression level of p-Akt, Akt, p-GSK-3β, GSK-3β, and β-catenin by Western blotting. The reduced level of p-Akt/Akt coinciding with the downregulation of p-GSK-3β/GSK-3β and β-catenin was clearly observed in the spheroids treated with 0.1–0.5 μM jorunnamycin A (Figure 3D). Surprisingly, jorunnamycin A at 0.05 μM also significantly decreased signaling in the Akt/GSK-3β/β-catenin pathway in CSCs spheroids (Figure 3E), though a minor alteration of stemness transcription factors is depicted (Figure 3A).

### 2.4. The Inhibitory Effect of Jorunnamycin A in Cscs of Various Lung Cancer Cells

The investigation of the inhibitory effect of jorunnamycin A in CSCs was further extended in human lung cancer H23 and A549 cells. Using LDA, the frequency of cancer spheroid-initiating cells in the lung cancer H23 and A549 populations are respectively depicted in Figure 4A and Figure 5A. The spheroid forming ability appears slightly less pronounced in lung cancer A549 cells, as evidenced by the smaller size of cancer colonies compared with H460 and H23 cells. Figure 4B and Figure 5B respectively indicate that nontoxic concentrations of jorunnamycin A (0.05–0.5 μM) significantly reduced the number of colonies at all cell densities (1–200 cells/well) in both H23 and A549 cells.

The alteration of anchorage-independent growth of CSC-enriched H23 and A549 spheroids after culture with 0.05–0.5 μM jorunnamycin A for 7 days is depicted in Figure 4C and Figure 5C, respectively. The reduction of colony size in both H23 and A549 spheroids was promptly observed after incubation of the spheroids with 0.5 µM jorunnamycin A for 1 day (Figure 4D and Figure 5D). Although anchorage-independent growth of CSC-enriched H23 spheroids was significantly suppressed only at the highest concentration (0.5 µM), jorunnamycin A at all nontoxic concentrations inhibited anchorage-independent growth of CSC-enriched A549 spheroids in a time- and dose-dependent manner, which was similarly observed in H460 cells. Furthermore, Figure 4E and Figure 5E sequentially demonstrate the alteration of the CD133low and CD133high populations in CSC-enriched spheroids of H23 and A549 cells after culture with jorunnamycin A for 72 h. The CD133high population was significantly reduced by jorunnamycin A at 0.5 μM in both the H23 and A549 spheroids, compared with the untreated CSC spheroids (Figure 4F and Figure 5F).

Since the stemness factors are known to be transcriptionally activated by Akt/GSK-3β/β-catenin signal [24,28], the modulation on stemness transcription factors was further confirmed in CSCs obtained from lung cancer H23 and A549 cells via RT-qPCR. Figure 6A,B respectively presents the significant reduction of Nanog, Oct-4, and Sox2 mRNA level in CSC-enriched H23 and A549 spheroids that were cultured with 0.5 µM jorunnamycin A for 24 h. Interestingly, jorunnamycin A at lower concentrations (0.05–0.1 µM) dramatically suppressed mRNA expression of these stemness transcription factors in CSC-enriched A549 spheroids. Although the incubation with 0.5 µM jorunnamycin A for 24 h obviously downregulated the related regulatory proteins, including those of p-GSK-3β/GSK-3β and β-catenin, there was no significant alteration of the p-Akt/Akt signal in either H23 (Figure 6C,D) or A549 CSC subpopulations (Figure 6E,F). The obtained results strongly support the role of jorunnamycin A in modulating CSC phenotypes in human lung cancer cells via the GSK-3β/β-catenin pathway.

The safety profile of jorunnamycin A on stemness features of normal lung stem cells was clarified in human lung epithelial BEAS-2B cells. Capability to form new colonies of healthy lung stem cells was assessed in LDA, as presented in Figure 7A. Despite containing mesenchymal stem cells [38], only a low level of colony formation was observed in BEAS-2B cells at a density of 200 cells/well. Intriguingly, the minor alteration of colony number was noticed in jorunnamycin A-treated and cisplatin-treated human lung epithelial BEAS-2B cells compared with the untreated control (Figure 7B). Three-dimensional (3D) spheroid formation was also performed as usual to investigate the self-renewal activity of normal lung stem cells [39]. Surprisingly, no significant alteration of stem cell-enriched BEAS-2B spheroids was observed after treatment with jorunnamycin A (0.05–0.5 µM) or 25 µM cisplatin (Figure 7C,D).

Since CD133 serves not only as a CSC protein marker but also a mediator of pluripotency in normal stem cells [40], jorunnamycin A-treated BEAS-2B spheroids were assessed for CD133 expression level. Flow cytometry analysis demonstrated that treatment with jorunnamycin A did not alter the %CD133high subpopulation in BEAS-2B spheroids when compared with the control group (Figure 7E,F). Indeed, the upregulated expression level of CD133 has been considered as a prognostic marker for lung tumor pathology [41,42]. Therefore, the present results suggest that jorunnamycin A selectively suppresses CSC phenotypes in various lung cancer cells without alteration of stemness features in healthy lung epithelial cells.

### 2.5. Jorunnamycin A Sensitizes Cisplatin-Induced Apoptosis in CSC-Enriched Spheroids

Although platinum-based chemotherapy is widely regarded as a primary option for lung cancer treatment, cisplatin resistance is a major problem that causes poor clinical outcomes [43]. To address the issues of drug resistance and eventual tumor recurrence [44], the co-administration of a natural product with the standard therapy of cisplatin has been regarded as a plausible option for achieving synergy [45]. CSCs are resistant to traditional chemotherapeutic agents [46] and the sensitization of chemo-resistant cells to cisplatin was evaluated after co-administration of jorunnamycin A and cisplatin to CSC-enriched spheroids. The suppressive effect of cisplatin (25 µM) on CSC-enriched spheroids is indicated in Figure 8A. Pretreatment with 0.5 µM jorunnamycin A for 24 h obviously sensitized H460 spheroids to cisplatin cytotoxicity. CSC-enriched H460 spheroids treated with cisplatin were evidently diminished in size by day 3 and nearly absent by day 5 in response to pre-incubation with 0.5 µM jorunnamycin A. Furthermore, there was a significant difference in the relative size of the spheroids subjected to cisplatin-only treatment versus the cisplatin-treated spheroids precultured with jorunnamycin A, which was evident within 3–7 days of incubation (Figure 8B).

Resistance to chemotherapy has been attributed to lung CSCs [5]. Thus, the CSC-targeted effect of cisplatin was also elucidated through the detection of the CD133high population in CSC-enriched spheroids. CSC-enriched spheroids were pretreated with jorunnamycin A (0.5 µM) for 24 h and further incubated with 25 µM cisplatin for 3 days, then were subjected to flow cytometry analysis. Treatment with either cisplatin (25 µM) or jorunnamycin A (0.5 µM) was able to reduce the %CD133-overexpressing cells in CSC-enriched H460 spheroids (Figure 8C). In comparison, a higher reduction of CD133high population was found in cisplatin-treated spheroids that were pre-incubated with jorunnamycin A, compared with the cisplatin-treated only group (Figure 8D), which points to a synergistic effect.

To confirm the chemosensitizing activity of jorunnamycin A in CSCs of human lung cancer cells, annexin V-FITC/propidium iodide (PI) staining and subsequent flow cytometry was performed to characterize mode of cell death [47]. Flow cytometry revealed that the untreated, jorunnamycin A-treated and cisplatin-treated CSC populations displayed similar distribution patterns of living cells (annexin V^−^/PI^−^) and early apoptosis (annexin V^+^/PI^−^) (Figure 8E). In contrast with the spheroids that received either cisplatin or jorunnamycin A treatment alone, an increased level of early apoptosis cells was found in CSC-enriched spheroids that were pretreated with jorunnamycin A (0.5 µM) for 24 h and further incubated with 25 µM cisplatin for another 24 h (Figure 8F). On the other hand, there was no modification of the %necrosis (annexin V^−^/PI^+^) in any of the treatments. These results strongly indicate that jorunnamycin A sensitized the CSCs of H460 spheroids to cisplatin-induced apoptosis. Indeed, the chemosensitizing effect of jorunnamycin A was also observed in CSCs spheroids obtained from lung cancer H23 (Figure 9A,B) and A549 cells (Figure 9C,D), which strongly indicate the anticancer activity of jorunnamycin A targeting of various lung CSCs.

### 2.6. Modulation of p53 and Bcl-2 Family Proteins in CSC-Enriched Spheroids Mediated by Jorunnamycin A

To investigate the underlying mechanisms involved in the chemosensitizing activity of jorunnamycin A in CSCs of human lung cancer cells, Western blot analysis was performed in the CSC-enriched H460 spheroids after culture with jorunnamycin A (0.05–0.5 µM) for 24 h. Figure 10A presents the expression levels of tumor suppressor p53 protein and related apoptosis-modulating proteins. Remarkably, p53 was upregulated in CSC-enriched spheroids treated with jorunnamycin A for 24 h in a dose-dependent manner. In addition, jorunnamycin A (0.05–0.5 µM) significantly reduced the level of Bcl-2, an anti-apoptotic protein, while the expression of Mcl-1 (myeloid cell leukemia 1) and BAX (Bcl-2-associated X) was not altered in response to the treatment with jorunnamycin A (Figure 10B). This information further confirms the apoptosis-regulating capacity of jorunnamycin A in CSCs of human lung cancer cells.

## 3. Discussion

The search for new strategies for improving therapeutic outcomes in lung cancer patients has led to great research interest in CSCs, a unique subpopulation in tumor tissue [7,24,48,49,50]. CSCs display self-renewal capacity to generate identical daughter cells, accounting for heterogeneity, therapeutic resistance, and other malignancies [51,52]. Cumulative evidence indicates that the modulating self-renewal pathways of CSCs are a promising strategy for cancer therapy [52,53,54] since these self-renewing and extremely tumorigenic CD133-overexpressing subpopulations have been clinically observed to produce poor clinical outcomes [55]. The outstanding anticancer potential of jorunnamycin A was indicated by the suppressive effect and chemosensitizing activity in CSCs derived from various human lung cancer cells, including H460 (p53 and KRas wild type), H23 (p53 and KRas mutant), and A549 (KRas mutant). The composition of the subpopulation that overexpresses CSC protein markers and exhibits self-renewal activity has been demonstrated in these lung cancer cells [17,36]. The tumor-initiating activity of CSCs isolated from H460 and A549 cells has been reported in an in vivo experiment [36]. Additionally, the selective suppression on CSCs of jorunnamycin A was supported with no alteration of stemness traits in normal lung epithelial BEAS-2B cells cultured with jorunnamycin A (Figure 7).

As an improvement in preclinical testing, 3D spheroids that mimic in vivo conditions and contain important tumor features, particularly drug resistance and stem-like phenotype, serve as a more robust and valuable model for in vitro screening for lung cancer treatments [34,46]. Culture under detachment condition used in 3D spheroid assays has been used to successfully stimulate and maintain the self-renewal capability of the CSC subpopulation in cancer cells [26,27,56,57]. This coincides with the disclosed results revealing that secondary spheroids from various human lung cancer cells obtained from 3D anchorage-independent culture were found to comprise approximately 80% CSCs that highly express CD133, a CSC protein marker [34,54] (Figure 2F, Figure 4F and Figure 5F). Moreover, the overexpression of Sox2, Nanog and Oct-4, the transcription factors mediating self-renewal and proliferation in CSCs [52], are notably expressed in the untreated spheroids. Collectively, the present results suggest that the 3D spheroids are enriched with CSCs. It is worth noting that the number of CSCs characterized as CD133high cancer cells was only 1–5% of total human lung cancer populations maintained under attachment culture (data not shown). Therefore, the reduction of both the relative size (Figure 2D, Figure 4D and Figure 5D) and the %highCD133 cells of CSC-enriched spheroids (Figure 2F, Figure 4F and Figure 5F) distinctly demonstrated the suppressive effect of jorunnamycin A on self-renewal in CSCs of human lung cancer cells.

Self-renewal activity is modulated through the collaboration of Oct-4 and Sox2 and the consequent transcription of the Nanog stemness gene [58,59]. Specifically, abnormal expression of Nanog accounts for CSC self-renewal and proliferation, while Oct-4 is highly expressed and more upregulated in response to cisplatin treatment in lung CSCs [52]. The role of Sox2 in regulating tumor development and maintenance of pluripotency in lung cancer has also been documented [60]. Moreover, co-expression of Oct-4 and Nanog is required for the induction of CSC properties and the enhancement of malignancy in lung adenocarcinoma [61]. On the contrary, the depletion of these stemness transcription factors decreases self-renewal activity and tumor formation in lung cancer [25,26,27,62]. Remarkably, downregulated mRNA levels of Nanog, Oct-4, and Sox2 transcription factor (Figure 3A and Figure 6A,B) in jorunnamycin A-treated CSC-enriched spheroids evidently confirmed the inhibitory effect on self-renewal in lung CSCs.

In lung CSCs, the Akt molecule not only promotes survival but also regulates the expression of self-renewal transcription factors [25,63,64,65,66]. The downregulation of stemness transcription factors is mediated through the Akt/GSK-3β/β-catenin cascade [24], which correlates with the suppressive effect of jorunnamycin A on these key proteins in the CSC-enriched spheroids obtained from H460 cells (Figure 3D). Although jorunnamycin A-mediated downregulation of Akt signaling has been previously reported in various human lung cancer cells [35], the investigation of CSC-related mechanisms demonstrated the suppression of p-Akt/Akt level only in CSC-enriched H460 spheroids but not in CSC subpopulations derived from H23 (Figure 6C,D) and A549 cells (Figure 6E,F). The low correlation between Akt signal and down-stream molecules presented in H23 and A549 spheroids might result from the fact that the alteration of an up-stream mediator should be detected at an earlier time point. However, the suppressive effect on the CSC phenotype (Figure 2, Figure 4 and Figure 5) and diminution of related stemness transcription factors (Figure 3A and Figure 6A,B) indicates the inhibitory role of jorunnamycin A in the GSK-3β/β-catenin pathway. Therefore, the present study reveals that jorunnamycin A downregulation of Nanog, Oct-4, and Sox2 might be mediated through the GSK-3β/β-catenin signal, although the direct targets of jorunnamycin A have not been thoroughly elucidated.

CSCs are also known as tumor-initiating cells since these cells are responsible for maintaining primary tumors as well as initiating secondary ones [67]. The restraint on this defining trait of CSCs was suggested with the marked diminution of cancer spheroid-initiating cells assessed via LDA in jorunnamycin A-treated lung cancer cell populations (Figure 2B, Figure 4B and Figure 5B). Tumor initiation capacity has been found to be facilitated by EMT, a dynamic process of converting epithelial cells into a mesenchymal phenotype [68,69]. The molecular link between EMT and CSC traits of self-renewal and tumor initiation has garnered much interest. Indeed, manipulating EMT is regarded as a potential strategy for targeting CSCs [70]. A previous study proposed that jorunnamycin A at nontoxic concentration (0.05–0.5 µM) stimulates detachment-induced cell death and suppresses anchorage-independent growth in human lung cancer cells by inhibiting EMT [35]. The barely detected level of cancer spheroid initiation in response to treatment with jorunnamycin A at a low concentration of 0.05 µM (Figure 2A, Figure 4A and Figure 5A) might result from the combined activity of sensitizing detached cell death, inhibiting EMT, and suppressing of the CSC phenotype. Nevertheless, the efficacy of jorunnamycin A at lower concentration (<0.05 µM) on tumor initiation in both in vitro and in vivo models should be further elucidated.

Although cisplatin or cis-diamminedichloroplatinum (II) has been widely used as a first-line platinum based-chemotherapy for lung cancer patients [71], eventual drug resistance and cancer relapse severely limit clinical benefit [72,73]. The subpopulation of CSCs in tumors are recognized contributors to cisplatin resistance in lung cancer [23,74,75]. The promising CSC-targeting activity of jorunnamycin A was demonstrated by its enhancement of cisplatin-induced apoptosis in CSC-enriched spheroids (Figure 8C,D). Pre-incubation for 24 h with jorunnamycin A (0.05–0.5 µM) activated p53 and subsequently downregulated anti-apoptosis Bcl-2 protein in CSC-enriched lung cancer cells (Figure 10). These results correspond with previous data showing that p53 is a key modulator for drug sensitization in CSCs [76] and a promoter of apoptosis through direct inhibition on Bcl-2 family proteins, including anti-apoptosis Bcl-2 [77]. Furthermore, Bcl-2 downregulation mediates apoptosis induction in chemo-resistant lung CSCs [78]. It should be noted that jorunnamycin A at nontoxic concentrations (0.05–0.5 µM) was previously reported to augment the level of BAX, a pro-apoptosis protein in human lung cancer cells [35]; however, the overexpression of BAX was not indicated in CSC-enriched spheroids treated with jorunnamycin A (Figure 10). Although sole treatment with either jorunnamycin A (0.5 µM) or cisplatin (25 µM) did not alter the %living cells, preincubation with jorunnamycin A prior to cisplatin treatment greatly triggered apoptosis (Figure 8E,F) and abolished the CD133high population in CSCs-enriched H460 spheroids (Figure 8C,D).

It is a fact that the inactivation of p53 and the consequent upregulation of downstream Bcl-2 protein contributes to chemotherapeutic resistance in both normal cancer and the CSC subpopulation. Although Bcl-2 inhibition has an established apoptosis-inducing effect in normal cancer cells, targeted Bcl-2 inhibition alone may insufficiently trigger apoptosis in CSCs [79]. Not only is the stem-like phenotype regulated by Nanog, a stemness transcription factor, but the apoptosis signal is also regulated by Nanog. It has been revealed that Nanog diminishes p53 expression [17,18,80]. Moreover, downregulation of Nanog and activation of p53 were found to efficiently improve chemotherapeutic response, especially in lung CSCs [81]. Corresponding with the results obtained in this study, the chemosensitizing effect of jorunnamycin A may result from the modulation on apoptosis-regulating proteins, including p53 and Bcl-2 mediated by stemness transcription factors.

## 4. Materials and Methods

### 4.1. Chemical Reagents

Jorunnamycin A was isolated from *Xestospongia* sp. that was collected by SCUBA diving at the vicinity of Si-Chang Island, Chonburi Province, Thailand, at a depth of 3–5 m with assistance from the Aquatic Recourses Research Institute, Chulalongkorn University, and permission from the Department of Fisheries, Ministry of Agriculture and Cooperatives, Thailand (0510.2/8234). The extraction, purification, and structure determination of naturally derived jorunnamycin A were performed based on the reported method [35,82]. Jorunnamycin A was obtained as an amorphous orange powder. The spectroscopic data of jorunnamycin A (Appendix A) was matched with the previous report [29]. The nuclear magnetic resonance (NMR) data of jorunnamycin A are presented in Table 2. For culture with lung cancer cells, jorunnamycin A was primarily dissolved in dimethyl sulfoxide (DMSO; EMD Millipore corporation, Billerica, MA, USA) then diluted to desired concentrations in culture medium. The final concentration of DMSO in culture medium was less than 0.5% (*v/v*).

Roswell Park Memorial Institute (RPMI) 1640 medium, Dulbecco’s modified Eagle’s medium (DMEM), fetal bovine serum (FBS), phosphate buffered saline (PBS), pH 7.4, and 0.25% trypsin containing 0.53 mM EDTA, l-glutamine, and penicillin/streptomycin solution were obtained from Gibco (Gaithersburg, MD, USA). Cisplatin, Hoechst33342, propidium iodide (PI), crystal violet solution (1% *w/v*), ethylenediaminetetraacetic acid (EDTA), and formaldehyde solution (37% *w/v*) were obtained from Sigma Chemical, Inc. (St. Louis, MO, USA). Bovine serum albumin (BSA) was purchased from EMD Millipore Corporation (Billerica, MA, USA). Annexin V/FITC-conjugated apoptosis detection kit and 3-(4,5-dimethylthiazol-2-yl)-2,5-diphenyltetrazolium bromide (MTT) were respectively procured from ImmunoTools (Friesoythe, Lower Saxony, Germany) and Life Technologies (Eugene, OR, USA). Primary antibody of Oct-4, Sox2, Nanog, β-catenin, Akt, p-Akt (Ser473), GSK-3β, p-GSK-3β (Ser9), p53, Mcl-1, Bcl-2, BAX, GAPDH, peroxidase-labeled specific secondary antibodies, and Alexa Fluor 488–conjugated secondary antibody were obtained from Cell Signaling Technology, Inc. (Denver, MA, USA). CD133 specific antibody was procured from US Biological life sciences (Salem, MA, USA). Bicinchoninic acid (BCA) protein assay kit and SuperSignal West Pico PLUS Chemiluminescent Substrate were sourced from Thermo Scientific (Rockford, IL, USA).

### 4.2. Cell Culture

Established human lung cancer cell lines (H460, H23 and A549) and human lung epithelial BEAS-2B cell line was obtained from the American Type Culture Collection (ATCC, Manassas, VA, USA). RPMI 1640 medium supplemented with 2 mmol/L l-glutamine, 10% (*v/v*) FBS, and 100 units/mL of penicillin/streptomycin was used for the culture of H460 and H23 cells. Meanwhile A549 and BEAS-2B cells were cultured in DMEM contained with 2 mmol/L l-glutamine, 10% (*v/v*) FBS, and 100 units/mL of penicillin/streptomycin. All cells were maintained in an incubator supplied with 5% CO_2_ at 37 °C until reaching approximately 80% to 90% confluence before further use in experiments.

### 4.3. Determination of Half-Maximal Inhibitory Concentration on Cell Viability (IC_50_) and Growth (IG_50_)

Cell viability was evaluated by MTT assay, which measures cellular capacity to reduce 3-(4,5-dimethylthiazol-2-yl)-2,5-diphenyltetrazolium bromide to purple formazan crystal by mitochondria dehydrogenase enzymes. After treatment with jorunnamycin A for 24 h, the cells, which were seeded at a density of 1 × 10^4^ cells/well in a 96-well plate, were then incubated with 0.4 mg/mL of MTT solution for 4 h at 37 °C away from light. MTT solution was then discarded and DMSO was added to dissolve the purple formazan crystals. The absorbance was determined by a microplate reader (Perkin Elmer, Waltham, MA, USA) at 570 nm. Percentage cell viability, which was calculated from the absorbance ratio of treatment to non-treated control cells, was used for determination of half-maximal inhibitory concentration (IC_50_).

Antiproliferative effect of jorunnamycin A was assessed through crystal violet assay. Cells were seeded at a density of 2 × 10^3^ cells/well in a 96-well plate and treated with nontoxic concentrations (0–0.5 µM) of jorunnamycin A for 72 h. At the indicated time point, the cells were washed with deionized water and fixed with 1% (*w/v*) formaldehyde for 30 min. Then, they were immersed in 0.05% (*w/v*) crystal violet solution for 30 min. After washing twice with deionized water and left air-dried, methanol (200 µL/well) was added to dissolve the crystal violet-stained biomass. The optical density (OD) was measured at 570 nm using a microplate reader (Perkin Elmer, Waltham, MA, USA). Percentage growth inhibition, which was represented after subtracting with OD of untreated control cells, was used for determination of half-maximal growth inhibitory concentration (IG_50_).

### 4.4. Limiting Dilution Assay

Limiting dilution assay (LDA) is used to measure the frequency of tumor initiating cells in cancer populations [83]. Human lung cancer cells were seeded into a 96-well ultralow attachment plate in gradually decreasing numbers from 200 to 1 cells/well in 200 µL of culture medium containing 1% (*v/v*) FBS with or without jorunnamycin A at nontoxic concentrations. After 14 days, the morphology and number of forming cancer colonies were observed under an inverted microscope (Nikon Ts2, Nikon, Japan).

### 4.5. Single Three-Dimensional (3D) Spheroid Formation

The enrichment of the CSC subpopulation in cancer cells was successfully performed through three-dimensional (3D) spheroid formation [57]. Briefly, human lung cancer cells (2.5 × 10^3^ cells/well) were maintained in culture medium supplemented with 1% (*v/v*) FBS under anchorage-independent condition in a 24-well ultralow attachment plate. After 7 days, the obtained primary CSC-enriched spheroids were resuspended into a single cell using 1 mM EDTA and seeded again into a 24-well ultralow attached plate (2.5 × 10^3^ cells/well). Secondary CSC-enriched spheroids were cultured for another 14 days before performing further experiments.

To determine the inhibitory effect on self-renewal [84], a single secondary CSC-enriched spheroid was isolated and treated with jorunnamycin A (0–0.5 µM). The alteration of spheroids after 0–7 days of jorunnamycin A treatment was observed under an inverted microscope (Nikon Ts2, Nikon, Japan) and presented as a relative value to spheroid size at day 0.

### 4.6. Determination of CD133-Overexpressing Cells in CSC-Enriched Spheroids via Flow Cytometry

Expression of CD133, a lung cancer stem cell marker, was evaluated by flow cytometry. After treatment for 3 days, jorunnamycin A-treated CSC-enriched secondary spheroids were collected and prepared into single cell suspension in PBS, pH 7.4. The cells were incubated with 0.5% (*w/v*) BSA in PBS for 30 min at 4 °C before probing with anti-CD133 antibody (US Biological life sciences, Salem, MA, USA: Cat no. 521102; Dilution 1:200) for 1 h at 4 °C. After washing the cell pellets with PBS, Alexa Fluor 488–conjugated secondary antibody (Cell Signaling Technology, Inc., Denver, MA, USA: Cat no. #4412S; Dilution 1:1000) was added and incubated for 30 min at 4 °C, protected from light. Fluorescence intensity was determined by flow cytometry (EMD Millipore, Billerica, MA, USA) using a 488 nm excitation beam and detection wavelength at 519 nm. Mean fluorescence intensity was quantified by guavaSoft version 3.2 software (EMD Millipore, Billerica, MA, USA).

### 4.7. Flow Cytometry Analysis of Annexin V-FITC/PI

Quantification of cell death in CSC-enriched spheroids was determined via flow cytometry analysis of annexin V-FITC/PI. After the indicated treatment, CSC-enriched secondary spheroids were collected and prepared into a single cell suspension in PBS, pH 7.4. Annexin V-FITC/PI staining was performed according to the manufacturer’s instructions. Briefly, cell pellets were collected and resuspended in 90 μL of binding buffer. The cell suspensions were stained with 5 μL of annexin V-FITC (1 μg/mL) and 5 μL of PI (2.5 μg/mL) for 20 min in a dark place. After adding of 400 μL of binding buffer, the cell samples were placed on ice for immediate analysis via a Guava easyCyte™ 5HT benchtop flow cytometer (EMD Millipore, Billerica, MA, USA) using guavaSoft version 3.2 software.

### 4.8. Reverse Transcription Quantitative Real-Time PCR (RT-qPCR)

Total mRNA was extracted from jorunnamycin A-treated CSC-enriched spheroids by using RevertAid First Strand cDNA Synthesis Kit (Thermo Scientific, Madison, WI, USA) according to the supplier’s protocol. Quantification of obtained cDNA was conducted by Thermo Scientific NanoDrop One microvolume UV-Vis Spectrophotometers (Thermo Scientific, Madison, WI, USA) at 260 nm. Primers specific to Nanog, Oct-4, Sox2, and glyceraldehyde-3phosphate dehydrogenase (GAPDH) were as follows:
● NanogForward: 5′-ACCAGTCCCAAAGGCAAACA-3′Reverse: 5′-TCTGCTGGAGGCTGAGGTAT-3′● Oct-4Forward: 5′-AAGCGATCAAGCAGCGACTA-3′Reverse: 5′-GAGACAGGGGGAAAGGCTTC-3′● Sox2Forward: 5′-ACATGAACGGCTGGAGCAA-3′Reverse: 5′-GTAGGACATGCTGTAGGTGGG-3′● GAPDHForward: 5′-GACCACAGTCCATGCCATCA-3′Reverse: 5′- CCGTTCAGCTCAGGGATGAC-3′.

Expression levels of transcription factor genes (Nanog, Oct-4, and Sox2) and housekeeping gene (GAPDH) in the CSC-enriched spheroids were analyzed by RT-qPCR using the CFX 96 Real-time PCR system (Bio-Rad, Hercules, CA, USA). One-step RT-qPCR reaction was carried using 50 ng of total cDNA using Luna Universal qPCR Master Mix (Bio-Rad, Hercules, CA, USA) with final volume of 20 μL per reaction. The initial denaturation step was performed at 95 °C for 3 min, followed by 40 cycles of denaturation at 95 °C for 5 sec and primer annealing at 57 °C for 30 s. The relative mRNA expression levels of the target genes were calculated using the comparative Cq values. The PCR products were normalized with the GAPDH gene as an internal control.

### 4.9. Western Blot Analysis

After treatment, CSC-enriched H460 spheroids were harvested and incubated in a RIPA lysis buffer (Merck, DM, Germany) containing 20 mM Tris-HCl (pH 7.5), 150 mM sodium chloride, 1 mM disodium EDTA, 1 mM EGTA, 1% NP-40, 1% sodium deoxycholate, 2.5 mM sodium pyrophosphate, 1 mM β-glycerophosphate, 1 mM sodium orthovanadate, 1 µg/mL leupeptin, and protease inhibitor cocktail (Sigma-Aldrich, St. Louis, MO, USA) for 40 min at 4 °C. The cell lysates were centrifuged at 12,000× *g* rpm (4 °C) for 15 min, then the supernatants were collected. Total protein contents were determined using BCA protein assay kit. An equal amount of protein of each sample was denatured by heating at 95 °C for 5 min with sample loading buffer and subsequently loaded onto a 10% SDS-PAGE. After separation, proteins were transferred onto 0.45 µM nitrocellulose membranes (Bio-Rad, Hercules, CA, USA). The transferred membranes were blocked for 1 h in 5% nonfat dry milk in TBST (25 mM Tris-HCl pH 7.5, 125 mM NaCl, and 0.05% Tween 20) and incubated with the appropriate primary antibodies at 4 °C overnight. Then, the membranes were washed with TBST (5 min × 3 times) and incubated with horseradish peroxidase-labeled isotype-specific secondary antibodies for 2 h at room temperature. Primary antibody of Oct-4 (Cat no. #2750S; Dilution 1:1000), Sox2 (Cat no. #3579S; Dilution 1:1000), Nanog (Cat no. #4903S; Dilution 1:2000), β-catenin (Cat no. #8480S; Dilution 1:1000), Akt (Cat no. #4691S; Dilution 1:1000), p-Akt (Ser473, Cat no. #4060S; Dilution 1:2000), GSK-3β (Cat no. #12456S; Dilution 1:1000), p-GSK-3β (Ser9, Cat no. #5558S; Dilution 1:1000), p53 (Cat no. #2527; Dilution 1:1000), Mcl-1 (Cat no. #94296S; Dilution 1:1000), Bcl-2 (Cat no. #4223S; Dilution 1:1000), BAX (Cat no. #5023S; Dilution 1:1000), GAPDH (Cat no. #5174S; Dilution 1:1000), and peroxidase-labeled specific secondary antibodies (Cat no. #7074S; Dilution 1:2000) for Western blot analysis were purchased from Cell Signaling Technology, Inc. (Denver, MA, USA). The immune complexes were detected by enhancement with chemiluminescence substrate and quantified by analyst/PC densitometric software (Bio-Rad, Hercules, CA, USA).

### 4.10. Statistical Analysis

All data are presented as means ± standard deviation (SD) from three independent experiments. The differences among the groups were evaluated by one-way analysis of variance (ANOVA), followed by Tukey HSD post-hoc test using SPSS version 22 (IBM Corp., Armonk, NY, USA). Statistical significance was defined as *p* < 0.05 for all tests.

## 5. Conclusions

This study reveals that jorunnamycin A selectively suppresses stem-like phenotypes in lung CSC-enriched spheroids through the inhibition of GSK-3β/β-catenin signal mediating downregulation of Nanog, Oct-4 and Sox2 transcription factors. Furthermore, the anticancer activity of jorunnamycin A targeting on CSCs is evidenced by the sensitizing effect on cisplatin-induced apoptosis in CSC-enriched lung cancer cells via upregulation of p53 tumor suppressor protein and decreased expression of anti-apoptosis Bcl-2 (Figure 11). These data support the further development of jorunnamycin A as an effective chemosensitizer for overcoming drug resistance and eradicating CSCs in lung cancer treatment.

## Figures and Tables

**Figure 1 marinedrugs-19-00261-f001:**
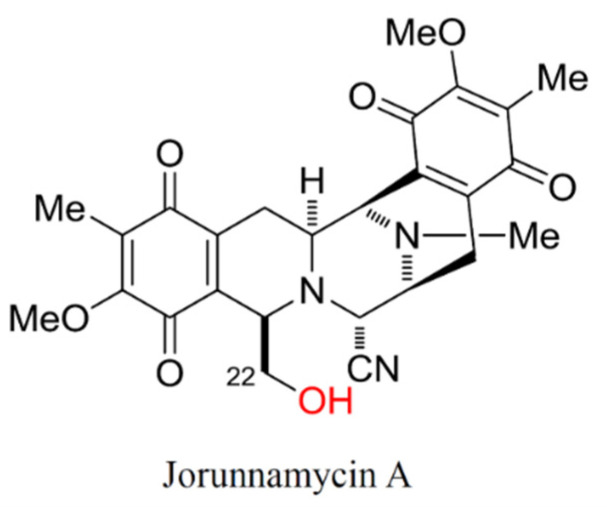
Chemical structure of jorunnamycin A. The hydroxyl group marked in red at C-22 ester side chain is the point of difference between jorunnamycin A and renieramycin M, another promising anticancer compound.

**Figure 2 marinedrugs-19-00261-f002:**
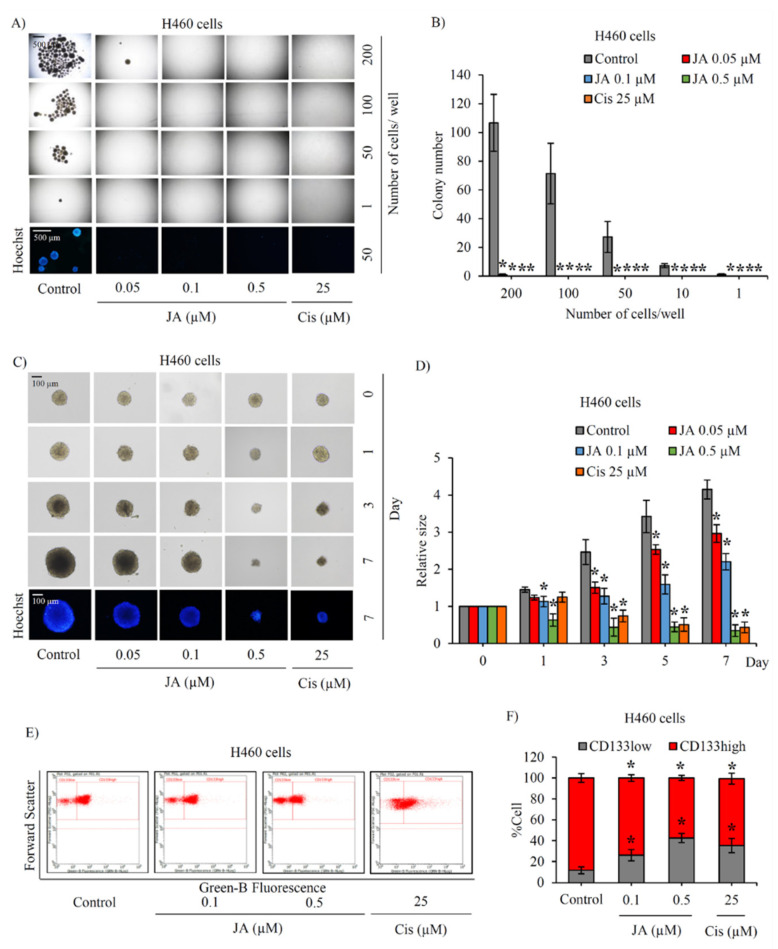
The inhibitory effect of jorunnamycin A on CSCs of lung cancer H460 cells. (**A**) Jorunnamycin A (JA) at 0.05–0.5 µM inhibited cancer spheroid-initiating capability of human lung cancer H460 cells, as evidenced by the suppression of cancer colony generation in the limiting dilution assay (LDA). At day 14 of treatment, spheroid images were captured by optical microscopy (4×), while bright blue fluorescence of Hoechst33342 was taken by fluorescence microscopy (10×). (**B**) The number of forming colonies at all cell densities was decreased in all nontoxic concentrations (0.05–0.5 µM) of JA. (**C**) Suppressive effect of JA in CSC-enriched lung cancer cells was evaluated by single three-dimensional (3D) CSC spheroid formation. At day 7 of treatment, all single 3D CSCs were visualized by optical microscopy (10×) and images reflecting the bright blue fluorescence of Hoechst33342 were taken by fluorescence microscopy (10×). (**D**) JA treatment exhibited a significant reduction of relative size of CSC-enriched spheroid from days 1–7. (**E**) Flow cytometry analysis of CD133 expression confirmed the alteration of CD133high population after treatment with JA (0.1–0.5 µM) for 3 days. (**F**) The proportion of CD133high cells in JA-treated population was lower compared with non-treated control. Cisplatin (Cis) at 25 µM was used as a positive control. The colony number (**B**) and relative size (**D**) were respectively analyzed in cancer colonies (**A**) and CSC-enriched spheroids (**C**). The %cell demonstrated in (**F**) was calculated from dot plots presented in (**E**). Data represent means ± SD of three independent experiments. ** p* < 0.05 versus non-treated control.

**Figure 3 marinedrugs-19-00261-f003:**
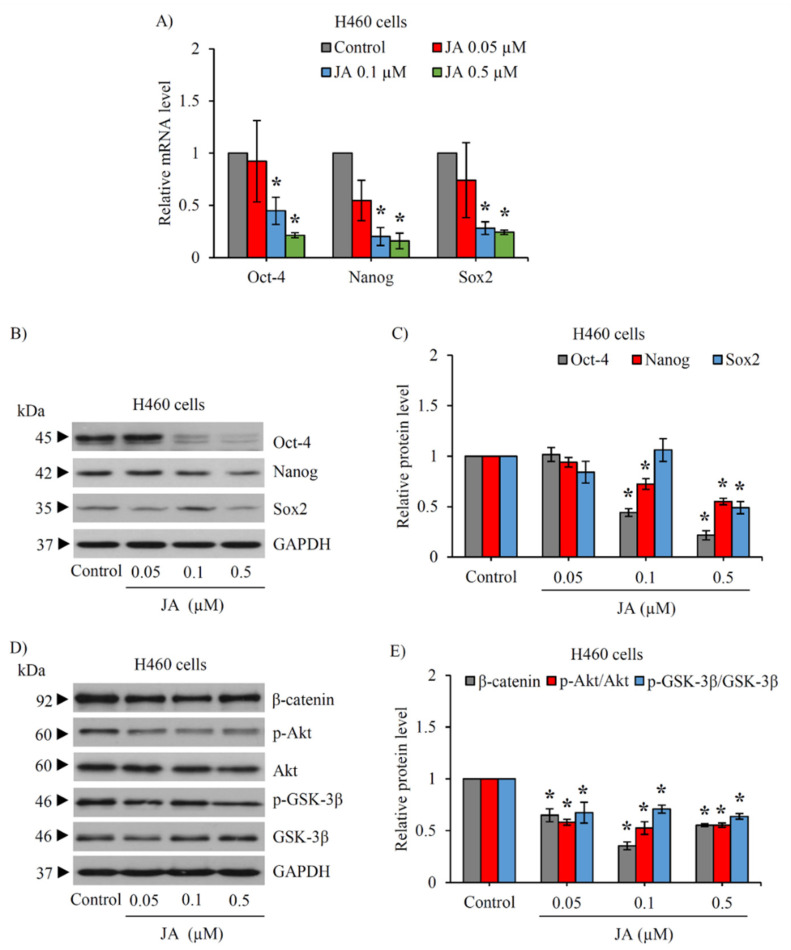
Jorunnamycin A downregulates stemness transcription factors and related proteins in CSC-enriched lung cancer H460 cells. The alteration of CSC self-renewal markers, Nanog, Oct-4, and Sox2, was determined in jorunnamycin A (JA)-treated CSC-enriched spheroids using (**A**) real-time RT-PCR and (**B**,**C**) Western blot analysis. GAPDH acted as an internal control. (**D**) Western blot analysis also revealed the downregulation of related up-stream proteins in CSC population of lung cancer H460 cells treated with JA for 24 h. (**E**) Noticeably, there was significant decrease of p-Akt/Akt, p-GSK-3β/GSK-3β, and β-catenin in JA-treated CSC-enriched spheroids. The relative protein level indicated in (**C**,**E**) was respectively analyzed from chemiluminescent signal detected in Western blotting presented in (**B**,**D**). Data represent means ± SD of three independent experiments. ** p* < 0.05 versus non-treated control.

**Figure 4 marinedrugs-19-00261-f004:**
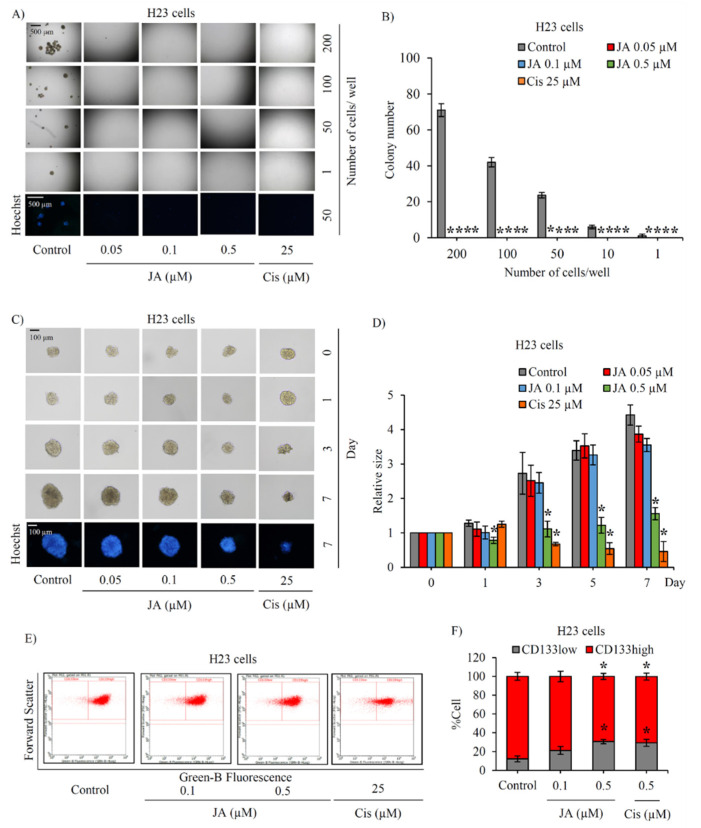
Jorunnamycin A suppresses stem-like phenotypes of lung cancer H23 cells. (**A**) The cancer spheroid-initiating cells of H23 cells were observed through limiting dilution assay (LDA). Jorunnamycin A (JA) inhibited the spheroid-forming capability of human lung cancer H23 cells in LDA for 14 days at all concentrations. At day 14 of treatment, optical microscopy (4×) was used to capture spheroid images, while fluorescence microscopy (10×) was used to visualize blue fluorescence of Hoechst 33342. (**B**) Decreased colony number was detected in all groups exposed to nontoxic concentrations (0.05–0.5 µM) of JA. (**C**) Single three-dimensional (3D) CSC spheroid formation was used to distinguish suppressive effect of JA in CSC-enriched lung cancer cells. At day 7 of treatment, all single 3D CSCs were visualized by optical microscopy (10×), and images depicting bright blue fluorescence of Hoechst 33342 were obtained by fluorescence microscopy (10×). (**D**) Treatment with JA (0.5 µM) exhibited a significant reduction of the relative size of CSC-enriched spheroids starting from days 1–7. (**E**,**F**) Flow cytometry analysis of CD133 expression confirmed the reduction of CD133high population after treatment with JA (0.5 µM) for 3 days. Cisplatin (Cis) at 25 µM was used as a positive control. The colony number (**B**) and relative size (**D**) were respectively analyzed from cancer colonies (**A**) and CSC-enriched spheroids (**C**). The %cell demonstrated in (**F**) was calculated from dot plots presented in (**E**). Data represent means ± SD of three independent experiments. ** p* < 0.05 versus non-treated control.

**Figure 5 marinedrugs-19-00261-f005:**
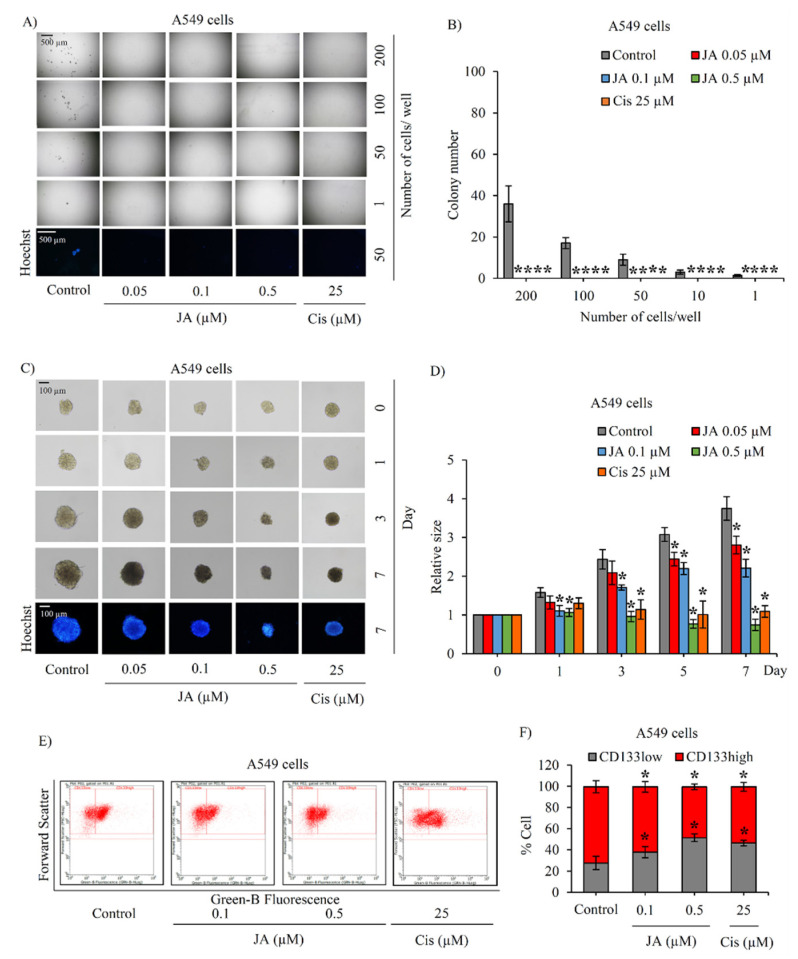
Suppressive effect of jorunnamycin A on stem-like phenotypes of lung cancer A549 cells. (**A**) Jorunnamycin A (JA) inhibited capability to generate cancer spheroid of human lung cancer A549 cells in limiting dilution assay (LDA) for 14 days at all concentrations. Optical microscopy (4×) was used to capture spheroid images, while fluorescence microscopy (10×) was used to take images with blue fluorescence from Hoechst33342 staining. (**B**) The number of forming colonies at all cell densities used was decreased in all groups exposed to nontoxic concentrations (0.05–0.5 µM) of JA. (**C**) Suppressive effect of JA in CSC-enriched A549 lung cancer cells was evaluated by single three-dimensional (3D) CSC spheroid formation. The enlargement of CSC spheroids was suppressed by JA (0.05–0.5 µM). At day 7 of treatment, all single 3D CSCs were photographed by optical microscopy (10×) and spheroid images depicting bright blue fluorescence of Hoechst33342 were obtained by fluorescence microscopy (10×). (**D**) Culture with JA (0.1–0.5 µM) illustrated the significant reduction of relative spheroid size of CSC-enriched lung cancer cells at days 1–7. (**E**) Flow cytometry analysis of CD133 expression confirmed the reduction of CD133high population after treatment with JA (0.1–0.5 µM) for 3 days. (**F**) JA at 0.1–0.5 µM diminished CD133high population in CSC spheroids compared with untreated control. Cisplatin (Cis) at 25 µM was used as a positive control. The colony number (**B**) and relative size (**D**) were respectively analyzed from cancer colonies (**A**) and CSC-enriched spheroids (**C**). The %cell demonstrated in (**F**) was calculated from dot plots presented in (**E**). Data represent means ± SD of three independent experiments. ** p* < 0.05 versus non-treated control.

**Figure 6 marinedrugs-19-00261-f006:**
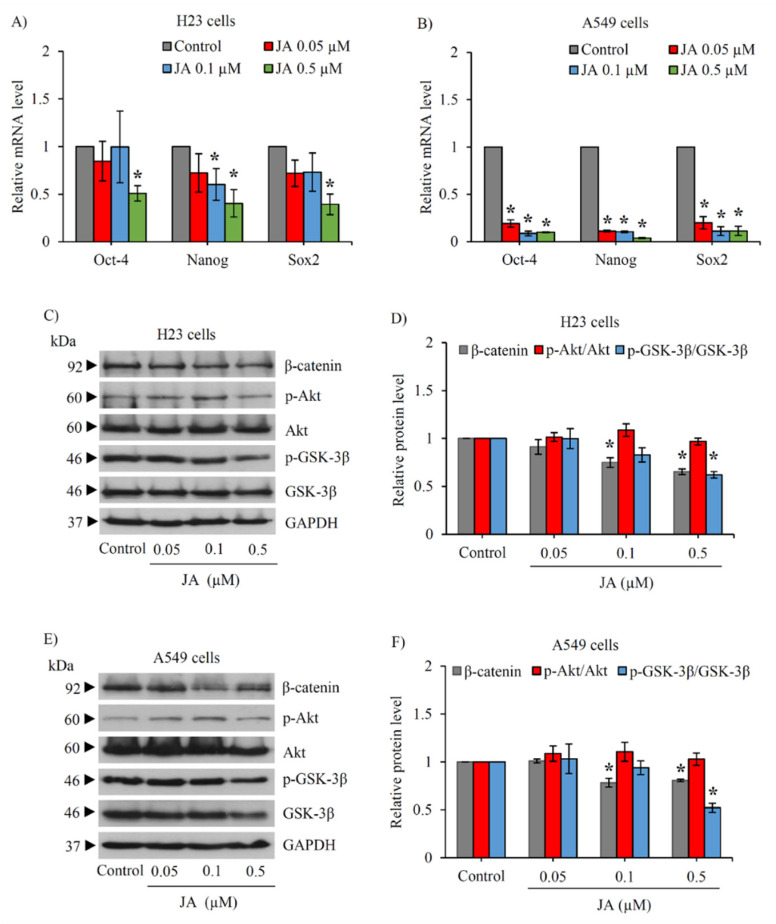
Jorunnamycin A downregulates stemness transcription factors and related proteins in various lung CSCs. The diminished mRNA levels of stemness transcription factors, including Nanog, Oct-4, and Sox2, determined by real-time RT-PCR is demonstrated in (**A**) CSC H23 and (**B**) CSC A549 spheroids cultured with 0.5 µM jorunnamycin A (JA) for 24 h. Western blot analysis revealed the downregulation of p-GSK-3β/GSK-3β and β-catenin in CSC-enriched population of (**C**,**D**) lung cancer H23 and (**E**,**F**) A549 cells treated with JA for 24 h. The relative protein level (**D**,**F**) was analyzed from chemiluminescent signal detected in Western blotting, as presented in (**C**,**E**). Data represent means ± SD of three independent experiments. ** p* < 0.05 versus non-treated control.

**Figure 7 marinedrugs-19-00261-f007:**
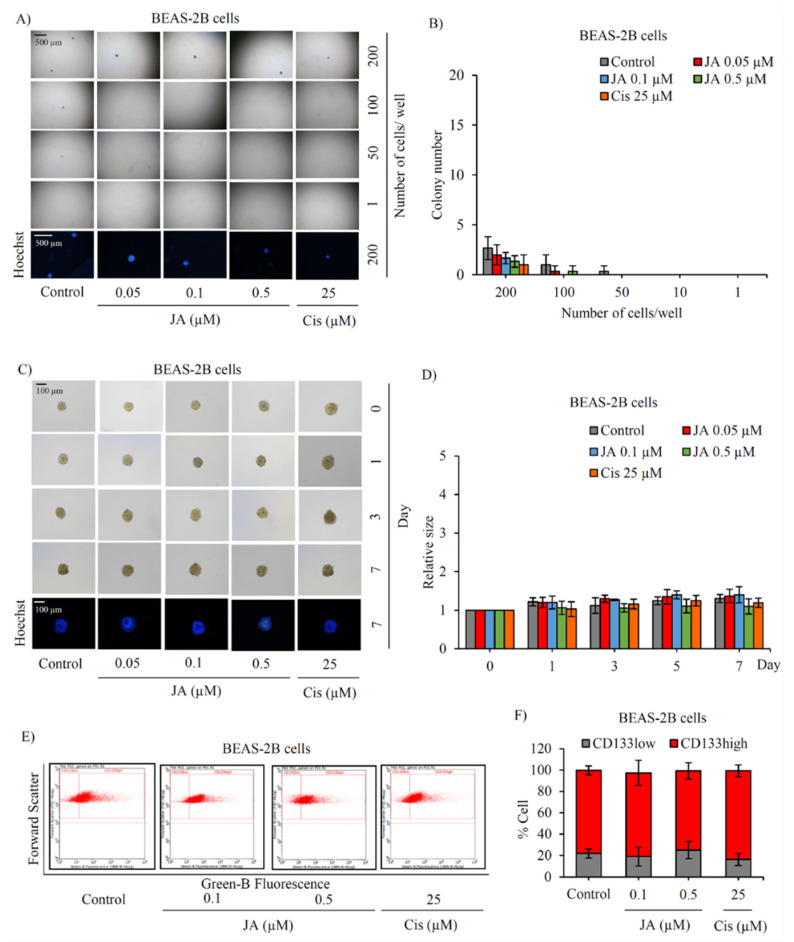
No alteration of stemness phenotypes in human lung epithelial cells cultured with jorunnamycin A. (**A**) Limiting dilution assay (LDA) revealed a low amount of colony formation obtained from normal lung epithelial BEAS-2B cells that were cultured in ultra-low attachment 96-well plate for 14 days. (**B**) There was no significant difference in the number of colonies generated from BEAS-2B cells incubated with 0.05–0.5 µM jorunnamycin A (JA), compared with untreated cells. Optical microscopy (4×) was used to capture spheroid images, while fluorescence microscopy (10×) was used to take images with blue fluorescence from Hoechst33342 staining. (**C**) Stemness features of BEAS-2B cells were also examined in single three-dimensional (3D) spheroid formation. After culture for 7 days, all single 3D spheroids were photographed by optical microscopy (10×) and spheroid images depicting bright blue fluorescence of Hoechst33342 were obtained by fluorescence microscopy (10×). Neither (**D**) relative spheroid size nor (**E**,**F**) %CD133-overexpressing cells in BEAS-2B spheroids were altered after treatment with JA (0.1–0.5 µM) when compared with untreated control. Cisplatin (Cis) at 25 µM was used as a positive control. The colony number (**B**) and relative size (**D**) were respectively analyzed from forming colonies (**A**) and stem cell-enriched spheroids (**C**). The %cell demonstrated in (**F**) was calculated from dot plots of flow cytometry analysis as presented in (**E**). Data represent means ± SD of three independent experiments.

**Figure 8 marinedrugs-19-00261-f008:**
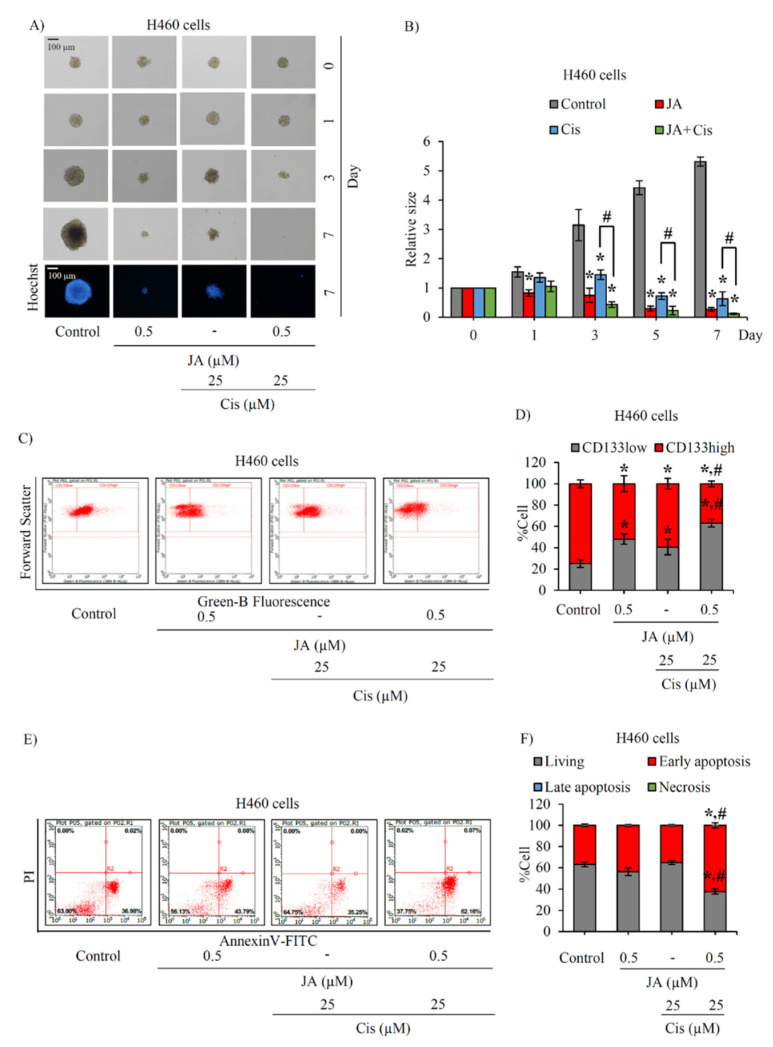
Sensitization of CSC-enriched lung cancer cells to cisplatin-induced apoptosis by jorunnamycin A. (**A**) Pre-incubation with jorunnamycin A (JA) followed by cisplatin (Cis) treatment dramatically suppressed CSC-enriched H460 spheroids cultured for 7 days. All single three-dimensional (3D) CSC spheroids were photographed by optical microscopy (10×), and spheroid images depicting bright blue fluorescence of Hoechst33342 were obtained by fluorescence microscopy (10×) at day 7 of treatment. (**B**) A greater decrease in the spheroid size relative to culture with cisplatin only was observed at days 3–7 in the combination treatment of 0.5 µM JA and 25 µM cisplatin. (**C**) Flow cytometry analysis reveals the decrease of CD133high population in JA-pretreated CSCs after incubation with cisplatin for 3 days. (**D**) The reduction of %CD133high population of CSC-enriched spheroids treated with cisplatin was evidenced in 24 h pretreatment with JA followed by 72 h cisplatin treatment. (**E**) Flow cytometry plots of CSC-enriched H460 spheroids co-stained with annexin V-FITC and propidium iodide (PI) indicate the increase of early apoptosis after the incubation with JA for 24 h prior to 24 h of cisplatin treatment. (**F**) A reduction of %living cells was significantly observed following cotreatment with JA and cisplatin in CSC-enriched population. The relative size indicated in (**B**) was analyzed from the morphology of CSC-enriched spheroids, as presented in (**A**). The %cell demonstrated in (**D**,**F**) was calculated from dot plots, as presented in (**C**,**E**), respectively. Data represent means ± SD of three independent experiments. ** p* < 0.05 versus non-treated control. ^#^ *p* < 0.05 versus only cisplatin-treated group.

**Figure 9 marinedrugs-19-00261-f009:**
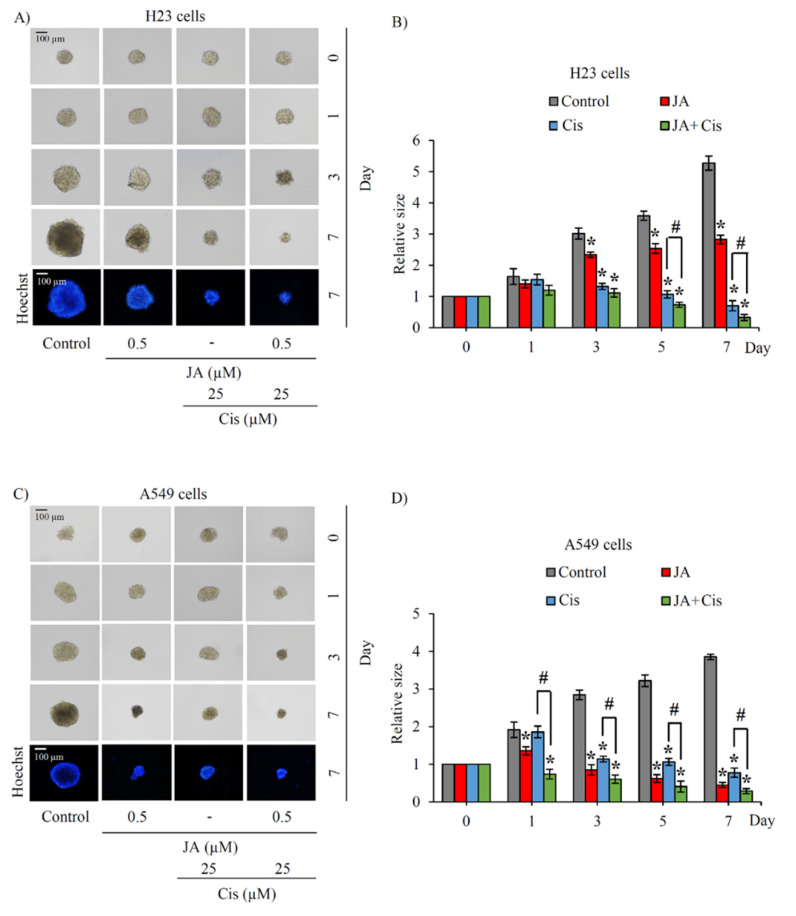
Cisplatin sensitizing effect of jorunnamycin A in various lung CSCs. CSC-enriched spheroids obtained from (**A**) H23 and (**C**) A549 lung cancer cells were precultured with nontoxic concentration (0.5 µM) of jorunnamycin A (JA) prior to treatment with 25 µM cisplatin (Cis) for 7 days. All single three-dimensional (3D) CSC spheroids were photographed by optical microscopy (10×), and spheroid images depicting bright blue fluorescence of Hoechst33342 were obtained by fluorescence microscopy (10×) at day 7 of treatment. Pretreatment with JA significantly diminished relative size of (**B**) H23 and (**D**) A549 CSC-enriched spheroids cultured with cisplatin. The relative size indicated in (**B**,**D**) was analyzed from the morphology of CSC-enriched spheroids, as presented in (**A**,**C**), respectively. Data represent means ± SD of three independent experiments. ** p* < 0.05 versus non-treated control. ^#^ *p* < 0.05 versus only cisplatin-treated group.

**Figure 10 marinedrugs-19-00261-f010:**
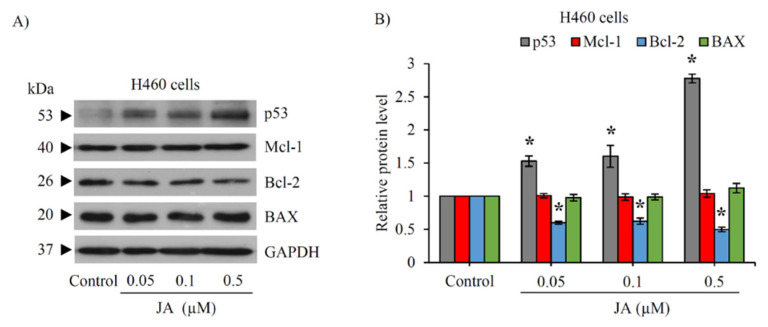
Alteration of apoptosis-related proteins in jorunnamycin A-treated lung CSCs. (**A**) Western blot analysis reveals the decline of anti-apoptosis Bcl-2 protein in CSC-enriched H460 spheroids incubated with 0.05–0.5 μM of jorunnamycin A (JA) for 24 h. (**B**) The relative protein levels were analyzed from the chemiluminescent signal detected in Western blotting, as presented in (**A**). JA obviously upregulated the expression level of p53, a tumor suppressor protein in CSC-enriched H460 cells, in a dose-dependent manner. Data represent means ± SD of three independent experiments. ** p* < 0.05 versus non-treated control.

**Figure 11 marinedrugs-19-00261-f011:**
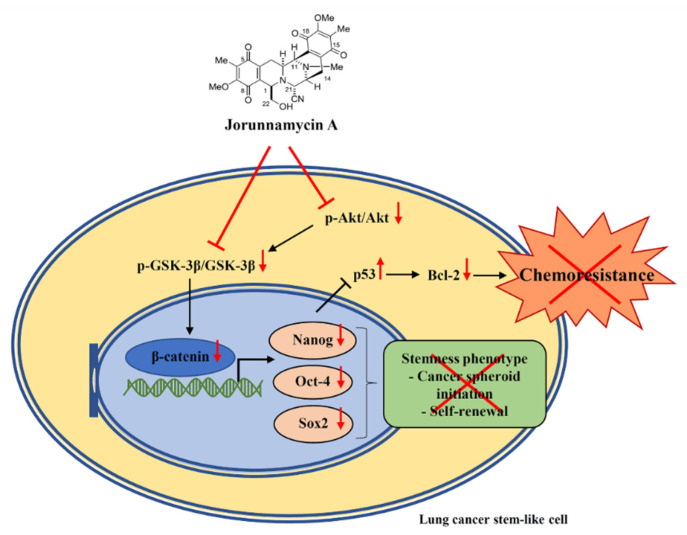
Schematic representation of jorunnamycin A-mediated suppression of CSC phenotype and sensitization of CSC-enriched lung cancer cells to cisplatin-induced apoptosis. The symbols and arrows in red color depict the effect of jorunnamycin A in CSCs of human lung cancer cells.

**Table 1 marinedrugs-19-00261-t001:** The half-maximal inhibitory concentration on cell viability (IC_50_) and proliferation (IG_50_) of jorunnamycin A in human lung cancer and normal lung epithelial cells.

Cell Type	IC_50_ (µM)	IG_50_ (µM)
H460	8.3 ± 2.6	0.27 ± 0.04
H23	2.0 ± 0.2	0.12 ± 0.01
A549	3.1 ± 0.3	0.23 ± 0.05
BEAS-2B	14.8 ± 0.6	0.65 ± 0.02

**Table 2 marinedrugs-19-00261-t002:** ^1^H NMR (400 MHz) and ^13^C NMR data (100 MHz) of jorunnamycin A in CDCl_3_ (δ in ppm).

No.	δ_C_, Type	δ_H_ (*J* in Hz)
1	58.0, CH	3.89, d (2.4)
3	54.3, CH	3.17, ddd (11.2, 2.6, 2.4)
4	25.3, CH_2_	2.92, dd (17.6, 2.4) 1.40, ddd (17.6, 11.2, 2.4)
5	185.4, C=O	-
6	128.9, C	-
7	155.5, C	-
8	181.4, C=O	-
9	136.0, C	-
10	141.7, C	-
11	54.1, CH	4.07, d (2.6)
12	41.5, N-CH_3_	2.30, s
13	54.4, CH	3.42, d (2.4)
14	21.6, CH_2_	2.82, dd (20.8, 7.2) 2.27, d (20.8)
15	186.2, C=O	-
16	128.6, C	-
17	155.4, C	-
18	182.2, C=O	-
19	135.6, C	-
20	141.7, C	-
21	59.0, CH	4.15, d (2.4)
22	64.1, CH_2_	3.71, dd (11.2, 3.2) 3.48, dd (11.2, 3.2)
6-CH_3_	8.7, CH_3_	1.93, s
7-OCH_3_	61.0, OCH_3_	3.98, s
16-CH_3_	8.7, CH_3_	1.93, s
17-OCH_3_	61.1, OCH_3_	4.03, s
21-CN	116.8, CN	-

## Data Availability

Appendix A are available online at www.mdpi.com/1660-3397/19/5/261/s1, Appendix A: NMR spectrum of jorunnamycin A.

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
