# Peer review of "Jorunnamycin A Suppresses Stem-Like Phenotypes and Sensitizes Cisplatin-Induced Apoptosis in Cancer Stem-Like Cell-Enriched Spheroids of Human Lung Cancer Cells"

_marinedrugs, 2021, doi:10.3390/md19050261_

Round 1

Reviewer 1 Report

Dear authors,

The article was very much improved.

I consider that the figures Fig 1,4,5,7 should be improved if some essential information are added on the figures (e.g., the tumor cell type...). If somebody is looking only to the figures, without reading the legend, there is not easy to understand the differences.

The same applies for Fig 6,8.

Author Response

Reviewer 1

The article was very much improved.

I consider that the figures Fig 1,4,5,7 should be improved if some essential information are added on the figures (e.g., the tumor cell type...). If somebody is looking only to the figures, without reading the legend, there is not easy to understand the differences.

The same applies for Fig 6,8.

Response: Reviewer’s comments and suggestion are appreciated. All figures were labeled with cell types.

Minor changes and re-formatting

  1. Minor corrections on grammar have been made.

*All changes in the revised manuscript are marked in red color.

Reviewer 2 Report

There are row numbers that overlap with tables in the pdf 108 - 112 and 575 - 576.

From what I see, the manuscript has come a long way compared to the previous submissions and respected most of the reviewers comments. Thus, I find it suitable for publication in its current form. (the misplaced numbers can be adjusted in the manuscript editing before publication if the editors will consider the manuscript suitable)

Author Response

Reviewer 2

There are row numbers that overlap with tables in the pdf 108 - 112 and 575 - 576.

From what I see, the manuscript has come a long way compared to the previous submissions and respected most of the reviewers comments. Thus, I find it suitable for publication in its current form. (the misplaced numbers can be adjusted in the manuscript editing before publication if the editors will consider the manuscript suitable)

Response: Reviewer’s comments and suggestion are appreciated. Due to use template of journal for preparing the manuscript, we cannot correct the overlapping of line number between the table.

Minor changes and re-formatting

  1. Minor corrections on grammar have been made.

*All changes in the revised manuscript are marked in red color.

Reviewer 3 Report

The authors presented an interesting research article aimed at evaluating the therapeutic properties of Jorunnamycin A, a sponge extract, in CSC lung cancer cells. For this purpose, the authors have performed several functional experiments analyzing stemness markers and morphological characteristics. The results presented are interesting and could improve the therapeutic options available for lung cancer patients. Below are reported some minor comments that the authors have to address before publication: 1) The authors should significantly reduce the length of the Discussion section. It is too verbose and there are redundant details; 2) How do the authors explain the inhibition of the tumor-initiating capability of lung cancer cells at a very low dose of Jorunnamycin A (0.05 uM). Have the authors performed experiments using low doses (<50 nM)? Please clarify these issues.  

tumor-initiating capability of human lung cancer

Author Response

Reviewer 3

The authors presented an interesting research article aimed at evaluating the therapeutic properties of Jorunnamycin A, a sponge extract, in CSC lung cancer cells. For this purpose, the authors have performed several functional experiments analyzing stemness markers and morphological characteristics. The results presented are interesting and could improve the therapeutic options available for lung cancer patients. Below are reported some minor comments that the authors have to address before publication:

1) The authors should significantly reduce the length of the Discussion section. It is too verbose and there are redundant details;

Response: The Discussion section was revised and corrected for more concise, better flow and shorter version

2) How do the authors explain the inhibition of the tumor-initiating capability of lung cancer cells at a very low dose of Jorunnamycin A (0.05 uM). Have the authors performed experiments using low doses (<50 nM)? Please clarify these issues.  tumor-initiating capability of human lung cancer

Response: The discussion about the inhibition on tumor-initiating capability of lung cancer cells at a very low dose of jorunnamycin A (0.05 uM) was added in the Discussion section as “A previous study proposed that jorunnamycin A at nontoxic concentration (0.05-0.5 µM) stimulates detachment-induced cell death and suppresses anchorage independent growth in human lung cancer cells via inhibiting EMT [35]. The barely detection of cancer spheroid initiation in response to treatment of jorunnamycin A at low concentration of 0.05 µM (Figure 2A, 4A and 5A) might result from the combined activity of sensitizing detached cell death, inhibiting EMT and suppression on CSC phenotype. Nevertheless, the efficacy of jorunnamycin A at lower concentration (< 0.05 uM) on tumor initiation in both in vitro and in vivo models should be further elucidated.”

References as they appear in manuscript:

  1. Ecoy, G.A.U.; Chamni, S.; Suwanborirux, K.; Chanvorachote, P.; Chaotham, C. Jorunnamycin A from Xes-tospongia sp. suppresses epithelial to mesenchymal transition and sensitizes anoikis in human lung cancer cells. J Nat Prod 2019, 82, 1861–1873, doi:10.1021/acs.jnatprod.9b00102.

Minor changes and re-formatting

  1. Minor corrections on grammar have been made.

*All changes in the revised manuscript are marked in red color.

Reviewer 4 Report

This manuscript contains some valuable data but needs to improve for publication.

1) Author needs to show the data about the expression of Nanog, Oct-4 and Sox2 by western blot, too.

2) Almost all experiments were done in CSC-enriched spheroids. So please reflect this to the title of this manuscript.

3) 'Tumour initiating' usually need to prove using mice. So please be careful about the use of tumour-initiating and stemness. 

4) Please discuss any differences in apoptosis in CSC and non-CSC cancer cells.

Author Response

Reviewer 4

This manuscript contains some valuable data but needs to improve for publication.

1) Author needs to show the data about the expression of Nanog, Oct-4 and Sox2 by western blot, too.

Response: As reviewer’s suggestion, we performed western blot analysis of Nanog, Oct-4 and Sox2 in CSC-enriched H460 spheroids treated with jorunnamycin A for 24 h. Although, the protein level of Nanog and Oct-4 was dose-dependently decreased in response to jorunnamycin A treatment, the significant reduction of Sox2 protein level was observed only in CSC H460 spheroids treated with 0.5 µM jorunnamycin A. The alteration of transcription factors obtained from western blotting seem to be less correlate to suppressive activity of jorunnamycin A. This might result from the limited sensitivity of western blot technique and the alteration of protein level should be detected at different time point. Additionally, Akt/GSK-3β/β-catenin signal regulates the transcription these stemness proteins which can be examined by RT-qPCR. Therefore, the alteration of stemness transcription factors in CSC-enriched spheroid of H23 and A549 cells was presented in RT-qPCR which was more correspondent with the targeting effect on CSCs of jorunnamycin A.  

      The western blot analysis of stemness transcription factors obtained from CSC-enriched H460 spheroids was added in Figure3B and C as well as in the Result section as following

- “The does-dependently decreased protein level of Nanog and Oct-4 detected via western blot analysis was also indicated in CSC-enriched spheroids cultured with jorunnamycin A for 24 h (Figure 3B). However, the significant reduction of Sox2 protein level was only observed in CSC H460 spheroids in response to treatment with 0.5 µM jorunnamycin A (Figure 3C).”

- “Due to be transcriptionally activated by Akt/GSK-3β/β-catenin signal [24, 28], the modulation on stemness transcription factors was further confirmed in CSCs obtained from lung cancer H23 and A549 cells via RT-qPCR.”

References as they appear in manuscript:

  1. Korkaya, H.; Paulson, A.; Charafe-Jauffret, E.; Ginestier, C.; Brown, M.; Dutcher, J.; Clouthier, S.G.; Wicha, M.S. Regulation of mammary stem/progenitor cells by PTEN/Akt/beta-catenin signaling. PLoS Biol 2009, 7, e1000121, doi:10.1371/journal.pbio.1000121
  2. Yong, X.; Tang, B.; Xiao, Y.F.; Xie, R.; Qin, Y.; Luo, G.; Hu, C.J.; Dong, H.; Yang, S.M. Helicobacter pylori upregulates Nanog and Oct4 via Wnt/beta-catenin signaling pathway to promote cancer stem cell-like properties in human gastric cancer. Cancer Lett 2016, 374, 292–303, doi:10.1016/j.canlet.2016.02.032.

2) Almost all experiments were done in CSC-enriched spheroids. So please reflect this to the title of this manuscript.

Response: The title of manuscript was changed to “Jorunnamycin A suppresses stem-like phenotypes and sensitizes cisplatin-induced apoptosis in cancer stem-like cell-enriched spheroids of human lung cancer cells”

3) 'Tumour initiating' usually need to prove using mice. So please be careful about the use of tumour-initiating and stemness.

Response: The term of “Tumor-initiating” presenting the results from in vitro LDA assay in this study was changed to “Cancer spheroid-initiating”.

                   For term of “Stemness” was changed to “Stem-like phenotype” in overall manuscript.

4) Please discuss any differences in apoptosis in CSC and non-CSC cancer cells.

Response: The discussion about the differences in regulating apoptosis in CSC and non-CSC cancer cells was added in the Discussion section as “It is the fact that inactivation of p53 and the consequent up-regulation of downstream Bcl-2 protein contribute to chemotherapeutic resistance in both normal cancer and CSC subpopulation. Unlike apoptosis inducing effect in normal cancer cells, the only inhibition on Bcl-2 might insufficiently trigger apoptosis in CSCs [79]. Not only stem-like phenotype but also apoptosis signal is regulated by Nanog, a stemness transcription factor. It has been revealed that Nanog diminishes p53 expression [17,18,80]. Moreover, downregulation of Nanog and activation of p53 were found to efficiently improve chemotherapeutic response, especially in lung CSCs [81]. Corresponding with the results obtained in this study, the chemosensitizing effect of jorunnamycin A may result from the modulation on apoptosis-regulating proteins including p53 and Bcl-2 mediated by stemness transcription factors.”

References as they appear in manuscript:

  1. Wang, Y.; Jiang, M.; Du, C.; Yu, Y.; Liu, Y.; Li, M.; Luo, F. Utilization of lung cancer cell lines for the study of lung cancer stem cells. Oncol Lett 2018, 15, 6791–6798, doi:10.3892/ol.2018.8265.
  2. Sarvi, S.; Mackinnon, A.C.; Avlonitis, N.; Bradley, M.; Rintoul, R.C.; Rassl, D.M.; Wang, W.; Forbes, S.J.; Gregory, C.D.; Sethi, T. CD133+ cancer stem-like cells in small cell lung cancer are highly tumorigenic and chemo-resistant but sensitive to a novel neuropeptide antagonist. Cancer Res 2014, 74, 1554–1565, doi:10.1158/0008-5472.CAN-13-1541.
  3. Tagscherer, K.E.; Fassl, A.; Campos, B.; Farhadi, M.; Kraemer, A.; Böck, B.C.; Macher-Goeppinger, S.; Radlwim-mer, B.; Wiestler, O.D.; Herold-Mende, C.; Roth, W. Apoptosis-based treatment of glioblastomas with ABT-737, a novel small molecule inhibitor of Bcl-2 family proteins. Oncogene 2008, 27, 6646-6656, doi: 10.1038/onc.2008.259.
  4. Golubovskaya, V.M. FAK and Nanog cross talk with p53 in cancer stem cells. Anticancer Agents Med Chem 2013, 13, 576–580, doi:10.2174/1871520611313040006.
  5. Chantarawong, W.; Chamni, S.; Suwanborirux, K.; Saito, N.; Chanvorachote, P. 5-O-acetyl-renieramycin T from blue sponge Xestospongia sp. induces lung cancer stem cell apoptosis. Mar Drugs 2019, 17, 109, doi:10.3390/md17020109.

Minor changes and re-formatting

  1. Minor corrections on grammar have been made.

*All changes in the revised manuscript are marked in red color.

Round 2

Reviewer 1 Report

Dear authors, the manuscript improved very much. However, in my version of the manuscript, there are the required changes in the body text, but I could not see the place where the figures were labeled with cell types.

Author Response

Reviewer 1

Dear authors, the manuscript improved very much. However, in my version of the manuscript, there are the required changes in the body text, but I could not see the place where the figures were labeled with cell types.

Response: For easily notify, the label of cell type was moved to the upper near A), B), C)......in each figure.

Reviewer 4 Report

All issues from me are cleared.

But the positions of labels of molecular weight and protein name in western blot need to be exchanged with each other. 

Author Response

Reviewer 4

All issues from me are cleared.

But the positions of labels of molecular weight and protein name in western blot need to be exchanged with each other. 

Response: The labels of transcription factors were rearranged in the same sequence of Oct-4, Nanog and Sox2 in the H460 cell results including RT-qPCR (Figure 3A), western blotting (Figure 3B) and relative protein level (Figure 3C). Additionally, the same label sequence was changed in RT-qPCR result of H23 (Figure 6A) and A549 cells (Figure 6B).

This manuscript is a resubmission of an earlier submission. The following is a list of the peer review reports and author responses from that submission.

Round 1

Reviewer 1 Report

The article is interesting. 

Abstract: there are no explanations for some abbreviations.

Introduction is very long. It should be more clear.

Results. should be revised.

Section 2.1. is very similar to the results presented in the first section of the article Ecoy et al. Jorunnamycin A from Xestospongia sp. Suppresses Epithelial to Mesenchymal Transition and Sensitizes Anoikis in Human Lung
Cancer Cells. Journal of natural products 2019, 82 (the article is cited at reference 33)

Section 2.7. has results from the article Ecoy et al. Journal of natural products 2019, 82.

Discussions. are clear.

Materials and Methods are clear.

Conclusion is clear.

References are up to date and reflect the major articles in the field.

Reviewer 2 Report

Please review English and make the article flow better.

The figures are too repetitive. Please consider to make them tell a story and to make them more aesthetically pleasing as I do not find dinamite plots (bar and standard deviation) appropriate. More than this, I do not recommend the use of different styles for the bars. Either keep one style or use different colors.

Overall, the methods used are rather standard, but the results might be of use for the scientific community.

Reviewer 3 Report

Sumkhemthong and colleagues presented a research article aimed at assessing the therapeutic potential of jorunnamycin A in lung cancer cells through the suppression of the stem-like phenotype of these cells. The manuscript is interesting and may pave the way to novel functional and clinical studies for the validation of this drug for the treatment of lung cancer patients. The manuscript is well written, however, there are some aspects that need to be clarified in order to improve the manuscript:

1) Please add the following information in a new table for a better visualization of data: “ 1H-NMR (CDCl3, 400 MHz)    in ppm: 4.15 (1H, d, J =2.4 Hz, 21-H), 4.07 (1H, d, J = 2.6 Hz, 11-H), 4.03 (3H, s, OCH3), 3.98 (3H, s, OCH3), 3.89 (1H, d, J = 2.4 Hz, 1-H), 3.71 (1H, dd, J = 11.2, 3.2 Hz, 22-Ha), 3.48 (1H, dd, J = 11.2, 3.2 Hz, 22-Hb), 3.42 (1H, d, J = 2.4 Hz, 13-H), 3.17 (1H, ddd, J = 11.2, 2.6, 2.4 Hz, 3-H), 2.92 (1H, dd, J = 17.6, 2.4 Hz, 4-Ha), 2.82 (1H, dd, J = 20.8, 7.2 Hz, 14-Ha), 2.30 (3H, s, NCH3), 2.27 (1H, d, J = 20.8 Hz, 14-Hb), 1.93 (6H, s, 6-CH3, 16-CH3), 1.40 (1H, ddd, J = 17.6, 11.2, 2.4 Hz, 4-Hb); 13C-NMR (CDCl3, 100 MHz)  in ppm: 186.2 (C-15), 185.4 (C-5), 182.2 (C-18), 181.4 (C-8), 155.5 (C-7), 155.4 (C-17), 141.7 (C-20), 141.7 (C-10), 141.3 (C-20), 136.0 (C-9), 135.6 (C-19), 128.9 (C-6), 128.6 (C-16), 116.8 (21-CN), 64.1 (C-22), 61.1 (OCH3), 61.0 (OCH3), 59.0 (C-21), 58.0 (C-1), 54.4 (C-13), 54.3 (C-3), 54.1 (C-11), 41.5 (NCH3), 25.3 (C-4), 21.6 (C-14), 8.7 (6-CH3), 8.7 (16-CH3).”;

2) In the material and methods section, the authors state “Human lung cancer H460, H23 and A549 cells were obtained from the American Type Culture Collection (ATCC, Manassas, VA, USA).”, however, the authors presented the results of the first round of experiments (MTT assay, PI3K/Akt pathway activation, etc.) only for H460 cells. Please clarify why the first experiments were not performed also for H23 and A549 cells;

3) The study is aimed at assessing the therapeutic potential of jorunnamycin A against cancer stem-like cells which are responsible for cisplatin therapeutic failure. However, the choice of cell lines used is questionable. Indeed, have these cells CSC properties? The authors should make this clearer. Did the authors assess the CSC capacity of untreated cells? Please, clarify;

4) In the subheading “Western blot analysis” please add the catalog numbers and suppliers of the antibodies used as well as the dilution used for the experiments;

5) In the Introduction or Discussion section, the authors should briefly describe the impact of immunotherapy in lung cancer with the adoption of immune checkpoint inhibitors (anti-PD-1, anti-PD-L1 e CTLA4). For this purpose, please see:

- 10.3390/cancers11101472

- 10.3389/fmed.2020.00090

- 10.3389/fphar.2018.01300

- 10.3332/ecancer.2017.787

6) Why Figure 8 and Figure 9 are in the Discussion section? Please move them to the Results section.

Reviewer 4 Report

It contains valuable data but needs to improve.

  1. In Figure 1. Why OH is red?
  2. In Figure 2D, I suggested that the author needs to emphasize the antiproliferative effects by redrawing a graph with Y as anti-proliferative effects.
  3. Why are the effects of JA  stronger in tumour initiating setting (Fig. 3A, B) compared with proliferation assay results (Fig. 2D).
  4. In Fig. 3B, D, F. it seems to be originated from FIg. 3A, C, E. If so, please indicate those things.  It seems to use the same data as separate figures.
  5. Page 6 line 167-168. The meaning is not clear. Please clarify the sentence.
  6. In Fig. 4C, it seems to be originated from FIg. 4B. If so, please indicate those things.  It seems to use the same data as separate figures. Please indicate that. These principles apply to other Figures. 
  7. In several bar graphs, please use different colours. 
  8. In the 2.6 part, please describe the results by emphasizing the meaning of sensitizing.
  9. In the discussion part, please discuss the direct target of JA. 
  10. Please discuss the role or significance of stem cell transcription factors in lung cancers.
  11. In the discussion part, please show the rationale for using lung cancer cell lines.

Round 2

Reviewer 1 Report

Thank you very much the answers to all questions or comments. The manuscript is improved.

I have some comments:

Personally, I consider that the introduction is quite long.

Results: Section 2.1.  I think that it is better to add a comment on the value of the experiment (repetability) that the experiment was done at another moment of time.

References: There are 2 articles corresponding to the same number in the references (e.g., 40, 44). These might be confusing. References should be revised in the text.

Reviewer 3 Report

The authors have addressed almost all of my previous comments. However, some concerns still persist mainly related to the choice of cell lines. Indeed, in their rebuttal letter, the authors state "Not only composing with high number of CSCs (~5% of total population) but also aggressive self-renewal activity and in vivo tumorigenicity...". Are 5% of CSCs sufficient to test the effects of Jorunnamycin A?

In addition, as stated in my previous comment 2, the first round of experiments should be performed in all the three cell lines selected. In this way, the authors will assess the reproducibility of Jorunnamycin A effects in lung cancer.

Reviewer 4 Report

All issues from me are cleared.

Author Response

Response The valuable suggestions from reviewer are appreciated.